# Comparison of Transforaminal Lumbar Interbody Fusion in the Ambulatory Surgery Center and Traditional Hospital Settings, Part 1: Multi-Center Assessment of Surgical Safety

**DOI:** 10.3390/jpm13020311

**Published:** 2023-02-10

**Authors:** Scott M. Schlesinger, Benjamin R. Gelber, Mark B. Gerber, Morgan P. Lorio, Jon E. Block

**Affiliations:** 1Legacy Spine & Neurological Specialists, 8201 Cantrell Road, Ste. 265, Little Rock, AR 72227, USA; 2Neurological and Spinal Surgery, Bryan Medical Center West, 2222 S. 16th Street, Ste. 305, Tower B, Lincoln, NE 68502, USA; 3Neuroscience and Spine Associates, 6101 Pine Ridge Road, Ste. 101, Naples, FL 34119, USA; 4Advanced Orthopedics, 499 E. Central Pkwy., Ste. 130, Altamonte Springs, FL 32701, USA; 5Independent Consultant, 2210 Jackson Street, Ste. 401, San Francisco, CA 94115, USA

**Keywords:** fusion, interbody cages, ambulatory surgery center, safety, TLIF, CPT ^®^ 22630, 22633

## Abstract

(1) Background: The technological advances achieved with minimally-invasive surgery have enabled procedures to be undertaken in outpatient settings, and there has been growing acceptance of performing minimally-invasive transforaminal interbody fusion (TLIF) in the ambulatory surgery center (ASC). The purposeof this study was to determine the comparative 30 day safety profile for patients treated with TLIF in the ASC versus the hospital setting. (2) Methods: This multi-center study retrospectively collected baseline characteristics, perioperative variables, and 30 day postoperative safety outcomes for patients having a TLIF using the VariLift^®^-LX expandable lumbar interbody fusion device. Outcomes were compared between patients undergoing TLIF in the ASC (*n* = 53) versus in the hospital (*n* = 114). (3) Results: Patients treated in-hospital were significantly older, frailer and more likely to have had previous spinal surgery than ASC patients. Preoperative back and leg pain scores were similar between study groups (median, 7). ASC patients had almost exclusively one-level procedures (98%) vs. 20% of hospital procedures involving two-levels (*p* = 0.004). Most procedures (>90%) employed a stand-alone device. The median length of stay for hospital patients was five times greater than for ASC patients (1.4 days vs. 0.3 days, *p* = 0.001). Emergency department visits, re-admissions and reoperations were rare whether the patients were managed in the traditional hospital setting or the ASC. (4) Conclusions: There were equivalent 30 day postoperative safety profiles for patients undergoing a minimally-invasive TLIF irrespective of surgical setting. For appropriately selected surgical candidates, the ASC offers a viable and attractive option for their TLIF procedure with the advantage of same-day discharge and at-home recovery.

## 1. Introduction

Managing spinal surgery patients in the ambulatory surgical center (ASC) has gained increasing attractiveness due to procedural efficiencies and cost advantages combined with comparable clinical results to those achieved in the traditional hospital setting [1]. This shift in patient care setting has been the direct result of substantial improvements in anesthesia and pain management protocols, perioperative safety with rare adverse event occurrence, and acceptable clinical outcomes [2,3]. Perhaps most importantly, patients express greater satisfaction undergoing surgery in the ASC which they find less intimidating than the traditional hospital setting in addition to the benefit of same-day discharge and recovery in the comfort of their own home [4].

The continued expansion in the range of spinal surgeries undertaken in the ASC has mirrored technical developments that support increasingly less invasive operative approaches that minimize the size of the surgical “footprint” [5,6]. Lumbar fusion procedures such as transforaminal lumbar interbody fusion (TLIF) have enjoyed increasing clinical adoption in the ASC that correlates with significant advances in minimally-invasive approaches and technologies, such as the use of stand-alone expandable cages [7]. Indeed, Heemskerk et al. [8] recently reported that patients treated with minimally-invasive TLIF have similar two-year clinical outcomes as patient having open TLIF. Correspondingly, findings have been reported demonstrating the feasibility, safety and clinical utility of performing the TLIF procedure in the ASC [9,10,11].

This multi-center study extends previously reported single-center findings [9], and provides procedural and 30 day safety outcomes for patients having a minimally-invasive TLIF procedure with the stand-alone VariLift^®^-LX expandable interbody fusion device in the ASC compared to the traditional hospital setting.

## 2. Materials and Methods

This multi-center, retrospective study was undertaken to collect a perioperative data set including baseline characteristics, intraoperative variables, and near-term (30 day) postoperative safety outcomes for patients undergoing TLIF with the VariLift^®^-LX expandable lumbar interbody fusion device (Wenzel Spine, Austin, TX, USA) (Figure 1). The purpose of this study was to determine the comparative 30 day safety profile for patients treated with TLIF in the ASC versus the hospital setting. Three spine surgeons at three separate clinical sites in the US participated in this study. A total of 167 patients were included at a per site ratio of 39:27:101 across the three surgeons.

Chart review captured patient background data including age, gender, body mass index (BMI), smoking status, primary diagnosis, functional status, Charlson Comorbidity Index (CCI) (0-7), American Society of Anesthesiologists (ASA) physical status classification (I-IV), and details of prior spine surgery. Patients were selected in reverse chronological order to identify all eligible cases for inclusion in the analysis. Intraoperative data included number and location of treated levels, use of supplemental fixation, blood loss, transfusion requirements, operative duration, complications, and length of stay. The 30 day postoperative outcomes tabulated the frequency and timing of emergency department visits, hospital and intensive care unit admissions, re-operations, and adverse events including infections. For this study, the data were stratified and compared by operative setting: ambulatory surgery center (ASC) or traditional hospital.

All patients had a one- or two-level TLIF with or without supplemental posterior instrumentation such as pedicle screws (CPT ^®^ 22630, 22633). In all cases, the interbody fusion utilized the VariLift^®^-LX expandable lumbar interbody fusion device (Figure 1) [12]. VariLift^®^-LX is a posterior stand-alone expandable lumbar interbody fusion device cleared by the FDA (K180822) for 1 or 2 levels, PLIF or TLIF, with or without supplemental fixation, and intended for use with autograft and/or allograft tissue. All fusion procedures were conducted under general anesthesia.

The minimally-invasive surgical arthrodesis technique and procedures have been detailed previously [9]. Briefly, after patient positioning, anatomical landmark identification, and incision, microscopic technique and fluoroscopic confirmation were used and neural decompression was performed by bilateral or unilateral laminectomy, discectomy and preparation of the disc space for the implantation of the expandable interbody fusion device. The appropriately sized interbody device (VariLift^®^-LX) was then placed with fluoroscopic guidance, expanded and filled with morselized autograft and/or allograft bone. In total, 152 (91%) cases received a stand-alone device. In the remaining 15 cases, additional fixation with pedicle screw instrumentation was included in conjunction with a posterior lateral fusion.

Patients had the drain removed between postoperative days one and three, typically returning on the day after surgery for postoperative drain removal. They were instructed to wear a lumbar brace for four to six months when up and ambulating. An individualized program of physical therapy was recommended for all patients, postoperatively, and consisted of cardiovascular exercise, soft-tissue mobilization, nerve mobilization, motor control and strengthening, and joint mobilization. Patients then had a follow-up visit at two weeks, six weeks, three months and six months postoperatively. Postoperative radiographs were obtained on the third and sixth months.

Univariate descriptive statistics such as means, medians and associated variability measures as well as frequency distributions were computed for all background and perioperative characteristics. All variables were compared statistically between study groups using the two-sample *t*-test (2-tailed) for continuous outcomes and Fisher’s exact test and the Mann–Whitney U test for categorical variables as appropriate. Postoperative visits at 24 h, as well as within 7 and 30 days were tabulated as frequencies for each study group.

## 3. Results

A total of 53 patients were treated in the ASC and, in 114 patients, the surgical procedure was undertaken in the traditional hospital setting. Comparative background characteristics for patients in both study groups are provided in Table 1. Patients treated in-hospital were significantly older, in worse physical health, and more likely to have undergone previous spinal surgery than ASC patients. However, preoperative back and/or leg pain severity scores were similar between study groups.

Table 2 shows a fairly similar magnitude and distribution of perioperative values between study groups. However, patients managed in the ASC had almost exclusively one-level procedures (98%), whereas almost 20% of procedures performed in-hospital involved two-levels (*p* = 0.004). There was no statistical difference between study groups in the use of supplemental fixation with most procedures (91% overall) utilizing the interbody cage as a stand-alone implant irrespective of operative setting. While median blood loss was significantly greater among ASC patients, the amount of loss in either setting (250 vs. 100 cc) was clinically negligible and similar to previous comparisons between surgeries undertaken in the ASC versus in-hospital [9].

The median length of stay for patients treated in the traditional hospital setting was approximately five times greater than for ASC patients (1.4 days vs. 0.3 days, *p* = 0.001). All patients treated in the ASC were discharged before midnight on the day of the procedure. Figure 2 illustrates comparative distributions in length of stay between study groups.

There was almost double the proportion of intraoperative complications in the traditional hospital group, although the difference between study groups did not reach statistical significance. Two in-hospital patients required a blood transfusion.

During the 30 day postoperative observation period, five ASC patients (9%) visited the emergency department: one for a fall without X-ray evidence of skeletal trauma, two for generalized pain and one with hypotension with medication resolution, and one with residual leg pain with readmission. In the traditional hospital group, there were six patients (5%) that visited the emergency department: one for urinary retention requiring catherization, one for drain removal, two for leg pain and numbness (one requiring readmission), and two for confusion requiring medication adjustment (one requiring readmission).

Hospital re-admissions were rare in both groups during the initial 30 days, postoperatively, with two (4%) occurring in the ASC group and six (5%) occurring in the traditional hospital group. One ASC patient (2%) required revision surgery with implant removal and instrumented multi-level arthrodesis. Four hospital patients (3.5%) had a re-operations including one revision with implant removal, two hematoma evacuations, and one thecal sac repair.

## 4. Discussion

The results of this multi-center study corroborate and extend previously published single-center findings showing a similar post-operative safety profile between commercially-insured patients having TLIF in the ASC versus those having the same procedure in the traditional hospital setting [9]. Additionally, these results expand the growing body of clinical evidence supporting the safety and utility of ASC-based surgery for appropriately selected patients undergoing minimally-invasive spinal fusion procedures [1]. To support this shift in patient care setting, Bovonratwet et al. [13], using a propensity-matched sample of over 30,000 lumbar spinal fusion cases from the American College of Surgeons National Surgical Quality Improvement Program (NSQIP) database, demonstrated similar 30 day rates of readmission and no significant difference in the occurrence of adverse events between cases treated as outpatients and those treated in-hospital.

The primary objective of this investigation was to amass and evaluate a data set of perioperative procedural characteristics and postoperative safety outcomes across multiple surgeons performing minimally-invasive TLIF using the stand-alone VariLift^®^-LX expandable interbody fusion device. This database allowed us to stratify the results based on patient care setting (i.e., ASC vs. in-hospital) in a cohort of cases all managed with the same surgical procedure and implant. This between-group comparative analysis demonstrated an approximate five-fold difference in the length of stay following surgery favoring procedures undertaken in the ASC (*p* = 0.001).

Patients undergoing TLIF in the hospital were significantly older with a commensurately greater number of comorbidities. These differences between groups were likely the result of two factors. First, all of patients in this study had their index surgery prior to the initiation of the 2020 Hospital Without Walls program that provided for Medicare beneficiaries to undergo TLIF in the ASC as a response to Covid-19 exigencies. Consequently, patients ≥65 years of age having TLIF were treated exclusively in the hospital, with younger commercially-insured patients having the option of being treated in the ASC. Second, as might be expected with a novel treatment protocol, patients treated in the ASC setting were likely selected judiciously by the surgical team based, in large part, on their overall health profile, the degree of spinal degenerative involvement, and an assessment of their risk of intraoperative adverse events that would prevent a same-day discharge. In this study, ASC patients were managed almost exclusively with a single-level TLIF using the expandable VariLift^®^-LX device standalone without supplemental posterior fixation. Indeed, other investigators have identified that this profile of a younger, healthier patient undergoing a single-level procedure is a robust predictor of earlier time to discharge, lower readmission rates and better 30 day safety outcomes following minimally-invasive lumbar fusion procedures [10,14].

The results of the current study underscore the surgical safety of minimally-invasive TLIF in general, irrespective of surgical setting. We found similarly rare frequencies between study groups for emergency department visits and hospital re-admissions, with one patient in each group requiring revision surgery with implant removal during the 30 day observation period. This study also confirmed the previous observation of substantially reduced perioperative blood loss associated with minimally-invasive TLIF regardless of setting [15,16,17,18]. Although there was a statistically significant difference in blood loss between patients treated in the ASC (250 cc) and the traditional hospital setting (100 cc), the average volumes were well within the expected range of reported values for minimally-invasive TLIF of 126 cc to 772 cc [15,16,17,18,19]. The average blood loss was also substantially less than would be encountered in an open TLIF procedure, which is routinely in excess of 1000 cc [20,21].

There is a limited but growing body of clinical evidence supporting the clinical utility of minimally-invasive TLIF using a standalone expandable cage for the treatment of lumbar spinal disorders [7,22,23]. This is due, in large part, to the lack of FDA-approved implants for this indication with the VariLift^®^ cage having this rare distinction [12]. Consequently, much of the published clinical data using this approach has employed this device [9,24,25]. Of note, Neely et al. [25] conducted a retrospective chart review of 470 VariLift^®^ standalone patients (642 treated levels) and reported a solid fusion rate of 94% at 4 years follow-up with little evidence of implant subsidence or migration.

## 5. Conclusions

In this multi-center study, we noted equivalent 30 day postoperative safety profiles for patients undergoing a minimally-invasive TLIF with the VariLift^®^ expandable cage irrespective of surgical setting. Emergency department visits, re-admissions, and reoperations were rare whether the patients were managed in the traditional hospital setting or the ASC. For appropriately selected surgical candidates, the ASC offers a viable and attractive option for their procedure with the advantage of same-day discharge and at-home recovery.

## Figures and Tables

**Figure 1 jpm-13-00311-f001:**
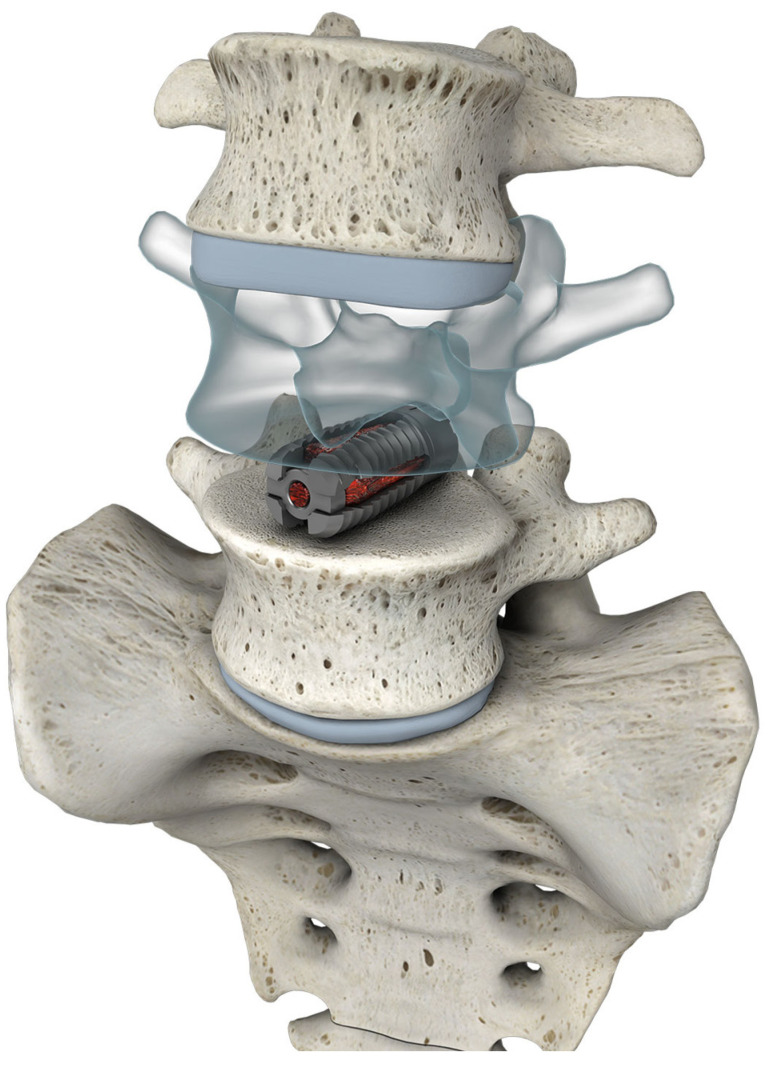
VariLift^®^-LX expandable stand-alone lumbar interbody fusion device (Wenzel Spine, Austin, TX, USA).

**Figure 2 jpm-13-00311-f002:**
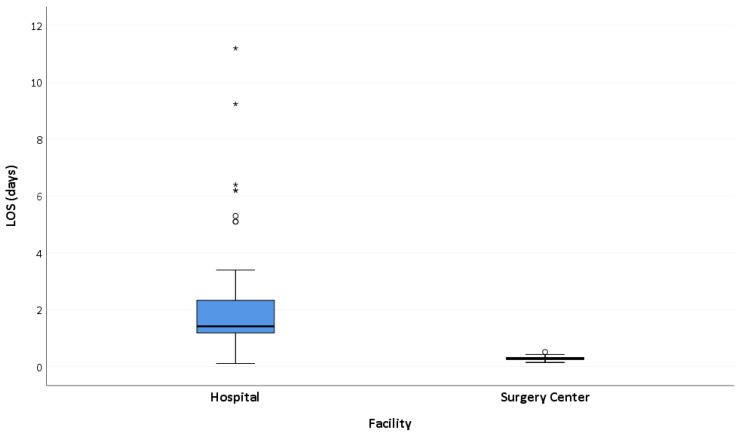
Box and whisker plot indicating length of stay by study group. The box indicates the upper and lower quartiles and the central line is the median. The points at the end of the “whiskers” are the 2.5% and 97.5% values. * Indicates outlying values.

**Table 1 jpm-13-00311-t001:** Background Characteristics.

Characteristic	ASC (*n* = 53)	In-Hospital (*n* = 114)	*p*-Value
Female, *n* (%)	29 (55)	60 (53)	0.87
Age, mean (SD) yrs	51 (9.8)	63 (13.0)	0.001
BMI, mean (SD) kg/m^2^	29 (4.5)	28 (5.1)	0.39
Smoker, *n* (%)	16 (31)	32 (28)	0.77
Primary Diagnosis, *n* (%)			
Degenerative Disc Disease	0 (0.0)	15 (13)	0.001
Spondylolisthesis	1 (2)	15 (13)	
Spinal Stenosis	52 (98)	81 (71)	
Other	0 (0.0)	3 (3)	
Prior Spine Surgery, *n* (%)	23 (43)	71 (62)	0.03
Adjacent-level Fusion, *n* (%)	6 (11)	20 (18)	0.36
Charlson Index, *n* (%)			
0	19 (36)	18 (16)	0.01
1	14 (26)	12 (11)	
2	16 (30)	23 (20)	
3	2 (4)	35 (31)	
4	1 (2)	20 (18)	
5	1 (2)	4 (4)	
6	0 (0)	2 (2)	
ASA Grade, *n* (%)			
I	2 (4)	0 (0)	0.003
II	36 (68)	59 (52)	
III	12 (23)	52 (46)	
IV	0 (0)	1 (1)	
Functional Status, *n* (%)			
Ambulatory	53 (100)	109 (96)	0.33
Wheelchair	0 (0)	5 (4)	
Leg Pain Score, median (range)	8 (1–8)	7 (1–9)	0.36
Back Pain Score, median (range)	5 (1–7)	7 (1–9)	0.49

**Table 2 jpm-13-00311-t002:** Perioperative Variables.

Variable	ASC (*n* = 53)	In-Hospital (*n* = 114)	*p*-Value
Treated Level, *n* (%)			
L2–3	0 (0)	7 (5)	0.003
L3–4	4 (7)	24 (18)	
L4–5	20 (37)	63 (47)	
L5–S1	30 (56)	39 (29)	
No. of Levels, *n* (%)			
1	52 (98)	95 (83)	0.004
2	1 (2)	19 (17)	
Supplemental Fixation, *n* (%)	3 (6)	12 (11)	0.79
Blood Loss, median (range) cc	250 (25–1100)	100 (19–1000)	0.001
Transfusion, *n* (%)	0 (0)	2 (2)	0.48
Operative Duration, mean (SD) hrs	2.1 (0.5)	1.8 (0.7)	0.008
Complications, *n* (%)	4 (8)	17 (15)	0.14
Length of Stay, median (SD) days	0.3 (0.15–0.52)	1.4 (0.11–11.2)	0.001

## Data Availability

The data presented in this study are available on request from the corresponding author. Individual participant data that underlie the results reported in this article will be made available (after de-identification) from 9 to 36 months after article publication.

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
