# Peer review of "Comparison of Transforaminal Lumbar Interbody Fusion in the Ambulatory Surgery Center and Traditional Hospital Settings, Part 1: Multi-Center Assessment of Surgical Safety"

_jpm, 2023, doi:10.3390/jpm13020311_

Round 1

Reviewer 1 Report

Schlesinger SM et al provided an interesting story of spinal surgery. Basically not relevant to the MS, I am interested to know the cost of VariLift® in TLIF between Traditional Hospital and APC setting. Is it feasible to convey these costs in (lower) middle income countries?

Author Response

We thank the reviewer for their thorough evaluation of our manuscript.

With respect to costs associated with TLIF in the ASC vs hospital, I would refer the reviewer to Gerling's article (reference #1) showing better cost-effectiveness for spine procedures performed in the ASC in general.  While we have not determined the comparative procedural costs for the patients included in this study, we are in the process of preparing a cost-analysis manuscript for the stand-alone VarilIft® device and that our preliminary estimates are $22,500 USD versus $32,600 USD for procedural costs for VariLift® and traditional PLIF, respectively.

Reviewer 2 Report

Thank you very much for the opportunity to review interesting research. I have a few comments:

1. please specify the purpose of the research both in the abstract and in the main body.

2. after the procedure, for the period of 6 months when the patients wore the brace, were they also recommended physiotherapy? if so what did it include?

3. please present the stages of patient recruitment together with the Flow Diagram.

Author Response

We thank the reviewer for their thorough evaluation of our manuscript.  We provide point-by-point responses below.

1)  A "purpose" statement has been added to the Abstract and Methods sections.

2)  An individualized program of physical therapy was recommend for all patients, postoperatively.  We have added text to this regard in the Methods section.

3)  As a retrospective chart review, this study employed a reverse chronological selection process for identifying eligible cases for inclusion in the analysis.  We have added clarifying text to the Methods section to distinguish this study from a prospective design that would require a patient flow chart.